# General and Treatment-Specific Outcomes with Osseointegrated Implants in Auricular, Nasal, and Orbital Prosthetic Reconstruction

**DOI:** 10.3390/cmtr18010016

**Published:** 2025-02-18

**Authors:** Morgan M. Sandelski, Deema Martini, Todd M. Kubon, Greg G. Gion, Amy L. Pittman

**Affiliations:** 1Department of Otolaryngology Head and Neck Surgery, Loyola University Medical Center, Maywood, IL 60153, USA; apittman@lumc.edu; 2Stritch School of Medicine, Loyola University Chicago, Maywood, IL 60153, USA; deemamartini@gmail.com; 3Medical Art Prosthetics, LLC, Madison, WI 53719, USA; tkubon@medicalartprosthetics.com; 4Medical Art Prosthetics, LLC, Brookfiled, IL 60513, USA; g.g.gion@sbcglobal.net

**Keywords:** craniofacial reconstruction, osseointegrated implants, head and neck reconstruction, radiation, cosmetic rehabilitation

## Abstract

Background: Osseointegrated implants outside of dental restoration remain an integral area of facial reconstruction in which more outcomes data is needed. We aimed to describe our 13-year experience using osseointegrated implants for orbital, nasal, and auricular reconstruction, looking at general outcomes, including radiated and surgically manipulated bone. Methods: This retrospective chart review covered demographics and outcomes from January 2008 to August 2021 in patients who underwent an orbital exenteration, partial or total rhinectomy, and partial or total auriculectomy with subsequent osseointegrated implant placement. We hypothesized radiation would increase the failure rate of implants and prostheses. Results: There were 79 implants placed in 27 patients, with over half of the patients requiring implants for reconstruction because of malignancy. The success rate was 86%. Complications were uncommon. Only 2 (7.4%) patients were unable to use their prosthesis. Prior radiation and surgery to the bone were associated with an increased risk of loss of implant (*p* = 0.008 and *p* = 0.007, respectively) but not associated with other complications or prosthesis non-viability. Conclusions: Osseointegrated implants are a reliable, permanent option for a realistic prosthesis. Radiation and prior surgery are significantly associated with an increased risk of implant failure but not associated with the inability to use the prosthesis. Regardless of prior treatments, bone-retained implants should be considered in facial reconstruction, especially after failing autologous repair or with concerns for cosmetic outcomes.

## 1. Introduction

Reconstruction of facial defects remains an important surgical challenge. The psychological effects of missing a part or having a significantly altered facial structure are devastating, in addition to the functional challenges, including difficulties wearing glasses and masks [1,2]. Autologous repair, namely of ears, had previously been the mainstay of reconstruction. However, this often requires multiple procedures, is technically challenging, and can lack a realistic and symmetric appearance. The importance of restoring the facial structure in terms of a patient’s psychological well-being cannot be underestimated.

Prostheses have grown in popularity over the years, given their realistic appearance and often simpler process compared to autologous repair [3,4]. Adhesives have been used to retain the prosthesis with struggling reliability in maintaining attachment and are associated with significant skin irritation [3,5]. Osseointegrated implants as a method to attach prostheses offer a solution to reconstruction that mitigates many of the challenges seen with autologous repair and adhesive attachment methods.

The field of osseointegrated implants has expanded beyond its origins with the work of Brånemark et al. in dental implants [6]. Early use in dental implants and bone-anchored hearing aids (BAHA) allowed for expansion into other areas. Osseointegrated implants for prosthetics provide a reliable, permanent option for reconstruction, largely known for dental restoration. Their use in orbital, nasal, and auricular reconstruction is less described. Whether the defect was obtained from a trauma, oncologic resection, or a congenital anomaly, a prosthesis provides a realistic and functional option for facial reconstruction.

A few studies have reviewed osseointegrated implants, but the data is limited for non-dental implants. Many of the larger volume studies review implantation for dental restoration [6,7,8,9,10,11,12]. Osseointegrated implants, as an option in facial reconstruction, have room to grow as a top option for auricular, nasal, and orbital defects of a variety of mechanisms of defects. The available literature often involves smaller cohorts with limited sample size, is outdated, and implanting these subsites remains controversial after radiation with contradicting information regarding the potentially negative impact of radiation [5,8,13,14,15,16,17,18,19,20]. We aim to provide more detail and enhance the pool of data on osseointegrated implants by providing data outcomes from a larger patient cohort. Throughout this paper, we detail our recent 13-year experience and pearls using osseointegrated implants for orbital, nasal, and auricular reconstruction. We look specifically at patients with prior radiation or surgical resection of the implanted bone. We hypothesize radiation will increase the failure rate of implants and prostheses.

## 2. Material and Methods

### 2.1. Study Design

After obtaining appropriate approval from Loyola University Medical Center’s institutional review board, a patient list was obtained from the International Classification of Disease (ICD) codes for partial and total auriculectomy, partial and total rhinectomy, and partial and total orbital exenteration from January 2008 to August 2021. Patients who came to at least one clinic visit after osseointegration were included in this study (n = 27). Osseointegration was determined clinically by the placing physician. Past medical history, surgical history, and the perioperative course were collected. All implants were placed by three experienced surgeons from the facial plastics and reconstruction or neurotology division. Areas of reconstruction were the ear, nose, and orbit. Patients required reconstruction for defects from malignancy, trauma, or congenital malformations.

### 2.2. Implant Procedure

Patients were either implanted with the Prior Generation or Vistafix 3 System model (Vistafix^R^ system—Cochlear Americas Corporation, Lone Tree, CO 80124, USA) in either a one-stage or two-stage procedure. Staged procedures were determined by the placing physician based on location and bone quality, most commonly electing for a two-stage procedure for a history of radiation therapy or surgical resection that might have altered the bony anatomy. The timing between stages of a two-stage procedure was determined by placing physicians depending on individual factors such as bone quality. All patients received perioperative surgical antibiotic prophylaxis of cefazolin and clindamycin if they had a penicillin allergy. Please note that in the writing of this manuscript, Cochlear has retired the Vistafix product line, and thus, the senior author now utilizes Southern Implants for bone-retained implants.

For one-stage procedures, the skin flap is raised to expose the bone. The bone is drilled, and the implants are placed. The skin flap is placed back over the implants, the skin over the implants is excised to expose the implant, and the healing abutment is secured to the implant. After osseointegration, the prosthesis can be placed on the abutments.

For two-stage procedures, the steps are the same; however, a cover is placed over the implant, and the first stage concludes as the skin flap is replaced. The period of osseointegration occurs between the first two stages. Thus, they are separated by 2–4 months. In the second stage, the skin flap is raised, exposing the implant. The cover is taken off, the skin replaced back down, the skin over the implants is excised to expose the implant, and the abutment and healing cap are placed.

### 2.3. Complications

Complications included skin overgrowth over the abutment, infection, and keloid or hypertrophic scarring. The timing of complications was divided into acute, defined as less than 3 months from surgery, and chronic, defined as greater than 3 months from surgery. Loss of implant was evaluated independently of the aforementioned complications.

### 2.4. Statistical Analysis

Statistical analysis was conducted with Graphpad Prism (version 10.1.1) using Fisher’s exact test.

## 3. Results

### 3.1. Patient Demographics

There were 27 patients that fit the criteria stated above. Table 1 details the patient demographics, implantation details, and osseointegration data. There were 12 (44%) patients who had radiation either pre- or post-implantation to the implanted bone and 14 (52%) patients who had prior surgery on the implanted bone. Between these groups, 11 (41%) patients fit both categories, leaving one patient who had radiation alone and three patients who had surgery alone as treatments for the implanted bone. Radiation data was available for six patients. All received an initial dose of 50–66 Gy, with two patients receiving subsequent doses of 50–60 Gy for recurrence. The remaining six patients did not have radiation information available. The average follow-up was 23 months.

### 3.2. Complications

Acute and chronic complications happened to seven patients in total (Table 2). Overall, 6 (22.2%) patients lost at least one implant. Implant survival was 86% (68/79). Three implants failed to osseointegrate, and the remaining eight were lost after osseointegration. Two patients (7.4%) were unable to use their prostheses because of the loss of implants (Table 3).

### 3.3. Radiation and Surgical Influence

There was no statistically significant difference in acute or chronic complications in patients who had received radiation or prior surgery to the implanted bone (Table 4). Prior surgery included lateral temporal bone resection (n = 10), partial maxillectomy (n = 3), and mastoidectomy (n = 1). Patients who had prior radiation or surgery to the implanted bone lost significantly more implants (*p* = 0.008 and *p* = 0.007, respectively). More patients who had prior treatment lost at least one implant compared to those who did not have radiation or surgery to the implanted bone; however, these results were not significant. The two patients who were unable to use their prosthesis had received radiation and had prior surgery on the implanted bone.

## 4. Discussion

### 4.1. Implant Location

Overall, our experience has shown acceptable success regardless of location, mechanism of injury, or comorbidities. There were 27 patients implanted with 79 implants with 86% of the implants surviving. Success rates in the literature range mainly from 75–100%, consistent with our findings [7,8,13]. Moore et al. had an overall success of 85%, but when looking specifically at Vistafix implants, they had a success of only 35%. Only 17 implants were placed, with nine orbital and eight nasal. The high failure rate was attributed to the Vistafix implants specifically being used more frequently in orbital reconstruction compared to the other brand, which was used in 143 implants for dental, orbital, nasal, and auricular reconstruction [8]. We only had one patient with implants for an orbital prosthesis, and there were no complications or loss of implants. However, the literature consistently cites lower success in orbital implants compared to other craniofacial sites [8,14,15]. While the exact mechanism is not known, it is proposed that the relatively diminished vascularity of the area, the thinner bone of the orbital rim, and the location being difficult to maintain proper hygiene contribute to higher failure rates in orbital implants [8]. One study found 100% success in implants for auricular reconstruction when vascularized tissue was placed after surgery in patients anticipating radiation, albeit with a very limited sample size [16]. This remains one method that must be considered to improve implant survival and should be especially considered in orbital reconstruction [17]. Additionally, the importance of adequate cleansing around the implants, regardless of location, should be emphasized to patients for maximal success. All in all, our experience with osseointegrated implants was positive and found to be reliable in multiple locations.

### 4.2. Radiation

Factors previously found to negatively impact implant survival include tobacco use and radiation [8]. A large portion of our cohort were never or prior smokers. One of the two patients who were unable to use their prosthesis was a current smoker, and both had prior radiation and surgery to the implanted bone. Radiation and surgery were not major factors in acute or chronic complications in our study but, unsurprisingly, were associated with implant failure. The significance of radiation is controversial in the literature. Moore et al. found a trend of increased survival in patients with various craniofacial implantations who did not have prior radiation in a cohort of 54 patients receiving 160 implants, but the difference was not statistically significant [8]. Similar results were seen in orbital reconstruction with 155 implants in 26 patients and with dentition in 102 implants in 20 patients [7,13]. De la Plata et al. found in their cohort of 169 dental implants in 30 patients that radiation therapy was significantly associated with implant failure. The success rate of the radiated group was 92.6% compared to 96.5% in the group that did not receive radiation, which brings into question the implications of this statistically significant difference. Both groups of patients had over 90% success, and although the radiation group had lower, the rate of implant survival remains excellent [9]. Our data showed a statistically significant difference in the number of implants lost in patients who had radiation and those who had surgery on the implanted bone. For the group with radiation, there was a 73.5% implant survival compared to 95.6% in the non-radiated group. More patients lost implants in the group with radiation, although this result was insignificant. Similar results were seen in the group with prior surgery to the bone, with 75% implant survival compared to 97.4% in the group that did not have surgically altered implanted bone. Hyperbaric oxygen (HBO) therapy is used as a preventative treatment in some studies, but data is inconsistent, with one randomized control trial showing no difference in implant survival [5,9,10,11,19]. There is no current overwhelming recommendation to use it as such, as many studies report similar success rates without HBO therapy.

Despite being associated with increased implant failure, radiation did not statistically increase the inability to use the prosthesis. It is important to recognize losing a single implant does not automatically render the prosthesis nonviable. Prostheses are often functional with fewer posts than implanted, sometimes requiring only one or two implants to successfully mount. In our study, only two patients (7%) were unable to use their prosthesis due to the loss of multiple implants. The remaining four patients who had lost at least one implant were able to use their prosthesis without issue with the remaining implants that successfully osseointegrated. One patient in our study had radiation following implant placement. This patient had three osseointegrated implants placed for nasal reconstruction. The only complication was skin overgrowth, and there was no loss of the implant. Regardless of prior or pending treatment, our data and the literature support acceptable success rates in implant survival and excellent rates of prosthesis viability.

### 4.3. Staged Procedure

A two-staged procedure should be considered in patients with prior radiation [19]. Per Vistafix system guidelines, a one-stage procedure is indicated for auricular prostheses and patients without poor wound healing risk factors [21]. In our patients, there were six two-staged procedures. One of these patients was unable to use their nasal prosthesis as all implants failed. However, the remaining five patients had 100% implant success for auricular reconstruction, with two of those patients with a history of radiation. There is minimal data on two-stage procedures in osseointegrated implants for auricular reconstruction; however, a limited number of patients have not had a loss of implant reported [3]. Of note is that our auricular prosthesis patients in the setting of malignancy all had lateral temporal bone resections. This surgery does not preclude the placement of osseointegrated implants; however, they should be placed with caution as the bone outside the resection can be thinner. It may also require creativity on the part of the anaplastologist in terms of more posteriorly placed implants that may not be within the field of the intended prosthesis. In our experience, we have found multidisciplinary visits with the surgeon, anaplastologist, and patient to allow for a robust conversation, including discussion of potential challenges specific to the patient’s anatomy and treatment history. Table 5 highlights our literature review regarding larger studies with extraoral implants with a special focus on radiation.

### 4.4. Local Reactions

Skin reactions can be a burdensome complication that can ultimately cause the patient to not use the prosthesis. Skin reactions were accounted for in the acute and chronic complications. Only seven patients (25%) had keloid formation, cellulitis, or skin overgrowth that ultimately did not impact prosthesis use. There was no difference in skin reactions between the radiated and non-radiated groups, which is consistent with other studies [3,14]. Besides proper hygiene and cleaning of the implant site, there are no effective treatments to prevent skin reactions. Prophylactic postoperative antibiotics have not been proven to decrease the incidence, which may indicate a noninfectious cause of these reactions [22].

### 4.5. Psychosocial Impact

Finally, the importance of reconstruction cannot be emphasized enough. The face is essential in self-identity, and significant psychosocial and functional deficits result from losing or altering an eye, ear, or nose (Figure 1a–c) [1,2]. Osseointegrated implants offer a reliable option to mount a realistic and functional prosthesis. Prostheses have a realistic nature that can lack autologous repair (Figure 2a–c). Five patients from our cohort had previous attempts at autologous repair and were unsatisfied with the results. All five patients had microtia. Osseointegrated implants with a prosthesis have the added benefit of often fewer procedures and a less complicated procedure than autologous repair [23]. A concern surrounding osseointegrated implants and subsequent prosthesis placement is cost. Ryan et al. found the costs of autologous repair and osseointegrated implantation with prosthesis to be comparable in unilateral procedures. In bilateral procedures, osseointegrated implants were significantly less expensive than autologous repair. A majority of the cost (70%) for osseointegrated implantation with prosthesis is from the cost of the prosthesis [4]. The average cost for the 23 patients from our cohort with financial information available is $6896. Insurance coverage varies, but we see Medicaid and several other private insurances cover a portion of the cost. Cost is an important discussion to have with patients seeking reconstruction and should involve discussion with anaplastologists. Ultimately, the cost can be similar with fewer procedures and reliable outcomes compared to autologous repair.

### 4.6. Limitations and Future Directions

There are a few important limitations to discuss. First, this is a retrospective chart review, which, by design, can lead to bias. Second, the power of the study is limited, which is higher powered; more significant results may have been seen. Lastly, there was a large overlap in patients from the radiation and surgically altered bone categories, limiting the ability to assess the effects of either independently. However, many patients seeking reconstruction for malignancy often present after radiation and surgery, so we believe this data and specific patient population are still relevant.

Future studies should aim to evaluate the effects of prior treatment in possibly a meta-analysis to better understand the effects. Additionally, patient satisfaction is important to consider in this process and is important to evaluate with further research.

## 5. Conclusions

Osseointegrated implants are continuing to rise as an option in reconstruction. Much of the available research details the use of dental implants. We have found from our own experience and literature review that the use in other craniofacial subsites is extremely promising regardless of prior treatment. Overall, we have had great success and hope our experience will highlight the versatility and reliability of osseointegrated implants as a great option in reconstruction.

## Figures and Tables

**Figure 1 cmtr-18-00016-f001:**
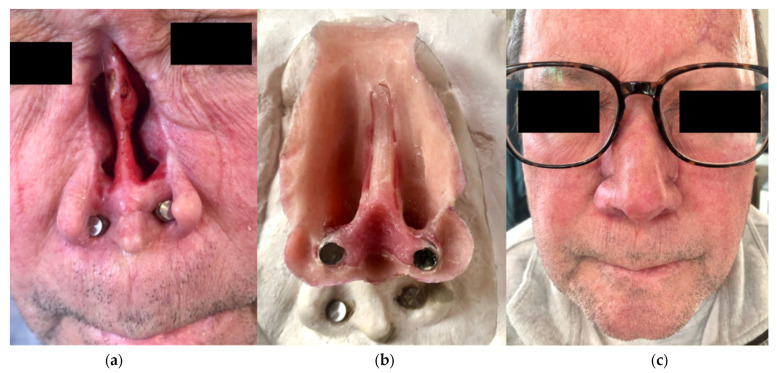
This patient had a total rhinectomy (**a**) as part of an oncologic resection and elected for osseointegrated implants (**b**) with a nasal prosthesis. His prosthesis allows him to easily wear his glasses (**c**).

**Figure 2 cmtr-18-00016-f002:**
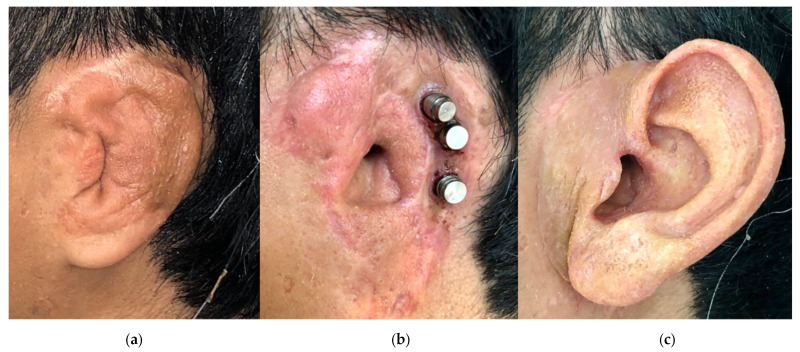
This patient had congenital microtia and previously had an autologous repair (**a**). He elected for osseointegrated implants (**b**) to support a prosthesis (**c**) for a more realistic shape.

**Table 1 cmtr-18-00016-t001:** Patient demographics are divided into categories based on the site of reconstruction with the number of patients (%) reported.

	EAR (n = 23)	NASAL (n = 3)	ORBIT (n = 1)
SEX (male)	18 (78%)	2 (67%)	1 (100%)
AVERAGE AGE (years)	46	65	16
DIABETES MELLITUS	3 (13%)	0	0
IMMUNOSUPPRESSED	2 (8.6%)	0	0
CURRENT SMOKER	3 (13%)	1 (33%)	0
MECHANISM OF INJURY			
Congenital	8 (35%)	0	0
Malignancy	12 (52%)	3 (100%)	0
Trauma	3 (13%)	0	1 (100%)
BONE SURGICALLY MANIPULATED	11 (48%)	3 (100%)	0
PRIOR RADIATION	9 (38%)	2 (67%)	0
RADIATION POST-IMPLANT	0	1 (33%)	0
STAGES OF PROCEDURE			
1	21 (91%)	0	0
2	2 (9%)	3 (100%)	1 (100%)
# IMPLANTS PLACED			
1	-	1 (33%)	-
2	-	-	1 (100%)
3	22 (96%)	2 (63%)	-
4	1 (4%)	-	-
Total implants placed	70	7	2
Vistafix system			
Prior generation	10 (43%)	0	0
Vistafix 3	13 (57%)	3 (100%)	1 (100%)
OSSEOINTEGRATION TIME (months)	2.96	8.67	4.5

**Table 2 cmtr-18-00016-t002:** Complications are divided into categories based on the site of reconstruction.

	EAR (n = 23)	NASAL (n = 3)	ORBIT (n = 1)
# PATIENTS			
Lost implant during osseointegration	1 (4%)	1 (33%)	0
Lost implant after osseointegration	4 (17%)	1 (33%)	0
Acute complications	3 (13%)	1 (33%)	0
Chronic complications	3 (13%)	0	0
Unable to use the prosthesis	1 (4%)	1 (33%)	0
	EAR (n = 70)	NASAL (n = 7)	ORBIT (n = 2)
# IMPLANTS			
Lost during osseointegration	1 (1.4%)	2 (28.6%)	0
Lost after osseointegration	7 (10%)	1 (14.3%)	0
Successful implants	62 (88.6%)	4 (57.1%)	2 (100%)

**Table 3 cmtr-18-00016-t003:** Demographics of the two patients who were unable to use their prostheses. Age is in years, and osseointegration is in months. DM—diabetes mellitus, RT—radiation therapy, chemo—chemotherapy.

Gender	Age	DM	**Immuno-** **Compromised**	Smoker	Reason for Implant	Stage	Implants Placed	Vistafix System	Osseo Integration	Location	Prior RT	Prior Chemo	Bone Altered
Male	68	No	No	Never	Cancer	One stage	3	Vistafix 3 system	3	Ear	Yes	Yes	Temporal bone resection
Male	55	No	No	Current	Cancer	Two stage	3	Vistafix 3 system	6	Nasal	Yes	No	Total rhinectomy, midline maxillectomy

**Table 4 cmtr-18-00016-t004:** Complications are divided by the history of radiation treatment and the history of surgery to the implanted bone. Values reported are the number of patients, except for the last category, which reports the number of implants. ** denoting level of significance.

	No Radiation	Radiation	*p*-Value		Bone Altered	Normal Bone	*p*-Value	
Acute complications	1	3			2	2		
None	14	9			12	11		
			*p* = 0.188				*p* = 1.00	
Chronic complications	3	0			0	3		
None	12	12			14	10		
			*p* = 0.231				*p* = 0.098	
Non-viable prosthesis	0	2			2	0		
Viable prosthesis	15	10			12	13		
			*p* = 0.188				*p* = 0.188	
Lost implant	2	4			5	1		
No implant loss	13	8			9	12		
			*p* = 0.358				*p* = 0.165	
# implants lost	2	9			10	1		
# implants retained	43	25			30	38		
			*p* = 0.008	**			*p* = 0.007	**

**Table 5 cmtr-18-00016-t005:** Literature review of implant success in patients with auricular, nasal, and orbital osseointegrated implants for prosthesis. Implants having received radiation were of special interest.

	# Patients	Location (# Patients if Multiple)	Radiation	# Implants	# Implants in Patients with RT	# Implants Failed/Failed in Patients with RT	Success Implants in Patients with RT
Benscooter et al. [16]	8	Auricular (7) Orbit (1)	4 (57.1%) -	25 2	15 (60%) -	1/1 0	93.3% -
Vijverberg et al. [3]	11	Auricular	3 (27.3%)	31	9 (29%)	0	100%
Wei et al. [14]	4	Orbit	2 (50%)	10	5 (50%)	1/0	100%
Moore et al. [8]	n/a	Auricular Orbit	n/a	2 36	2 (100%) 36 (100%)	0 8/8	100% 77.8%
Korfage et al. [5]	28	Nasal	20 (71.4%)	56	40 (71.4%)	2/1	97.5%
Vitomer et al. [20]	26	Nasal (15) Orbit (15)	4 (36.4%) 6 (40%)	28 38	11 (39.3%) 13 (34.2%)	2/1 4/4	90.9% 69.2%

n/a—not applicable.

## Data Availability

The raw data supporting the conclusions of this article will be made available by the authors on request.

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
