# Peer review of "General and Treatment-Specific Outcomes with Osseointegrated Implants in Auricular, Nasal, and Orbital Prosthetic Reconstruction"

_1943-3883, 2025, doi:10.3390/cmtr18010016_

Round 1
Reviewer 1 Report
Comments and Suggestions for Authors
Thank you for the opportunity to review this article.
It shows a retrospective single-center evaluation of the success using extraoral implants for anaplastology, suiting the general interest and topic of the journal.
Nevertheless, there are some minor concerns.
The abstract is clear and provides all relevant information needed.
The introduction gives an overview of the literature focusing the reconstruction of parts of the face and its problems. Nevertheless, there should be more information about extraoral implants and their outcome. Some information is provided in Table 5, which has no reference within the manuscript.
The methods are explained properly. Regarding the description of the perioperative management, some information are missing:
- Did the patients receive antibiotics as infection prophylaxes? If yes, what antibiotic has been chosen? And if their was a difference with in the patients, please put this fact also into consideration about occurrence of complications.
The results are structured clearly and provide all information needed.
The discussion shows an appropriate interpretation of the results.
The conclusion summarizes the whole manuscript in a short way.
I am curious about your answers and would be pleased to re-review your paper.
Author Response
Comment 1: The introduction gives an overview of the literature focusing the reconstruction of parts of the face and its problems. Nevertheless, there should be more information about extraoral implants and their outcome. Some information is provided in Table 5, which has no reference within the manuscript.
Response: The information is limited in the data, which is why we feel our study can contribute nicely. Most of our information is in the results. I did add the citation of table 5 in the discussion section at the end of the paragraph regarding radiation.
Comment 2: Did the patients receive antibiotics as infection prophylaxes? If yes, what antibiotic has been chosen? And if their was a difference with in the patients, please put this fact also into consideration about occurrence of complications.
Response: I have added the information regarding antibiotics. In short, every patient received periooperative antibiotic prophylaxis.
Reviewer 2 Report
Comments and Suggestions for Authors
Dear Authors,
This study is a study on the short-medium term survival of prostheses used in facial defects. The data of these studies vary according to the total surgeries performed in the relevant clinic, the type of surgery performed, and regionally common cancers or cases. Therefore, it may be difficult to find sufficient cases over many years. There are some limitations in terms of numbers in this study. It is observed that the results in this study are consistent with the literature. It was appropriate for clinicians to share their experiences. My suggestion would be to add the pathologies of the primary tissue loss cause. I believe that adding a table containing information such as tumor origin and type, whether or not it has received radiotherapy, or if there is trauma, its size and shape would be appropriate in terms of containing explanatory information.
Best regards.
Author Response
Comment 1: My suggestion would be to add the pathologies of the primary tissue loss cause. I believe that adding a table containing information such as tumor origin and type, whether or not it has received radiotherapy, or if there is trauma, its size and shape would be appropriate in terms of containing explanatory information.
Response: Thank you for this comment. In table 1 we discuss etiology of congenital, trauma, or malignancy. Further differentiation would lead to significant line additions to the table without much further information for malignancy. In regards to trauma, most of the traumas were seen after healing outside of one. We also have radiation therapy data in table 1.
Reviewer 3 Report
Comments and Suggestions for Authors
The manuscript ”General and Treatment-Specific Outcomes with Osseointegrated Implants in Auricular, Nasal, and Orbital Prosthetic Reconstruction” presents a retrospective research article focusing on the clinical outcomes of osseointegrated implants used in auricular, nasal, and orbital prosthetic reconstructions. It presents clinical data collected over 13 years, analyzes treatment-specific and overall outcomes, and discusses factors influencing implant success rates. The manuscript provides valuable information on the outcomes of osseointegrated implants for craniofacial reconstruction, an area with limited existing literature. The study stands out for its focus on non-dental applications and its robust data record spanning 13 years. However, there are areas for improvement regarding clarity, structure, and depth of analysis.
A. Abstract - remove ”study design”; separate the purpose of the study from the methods. State the retrospective nature of the study early in the abstract. Include sample size and time frame concisely. Highlight key numerical results with concise language, such as implant success rates. Explicitly state the significance of results related to radiation and surgery.
B. Introduction - reinforce this section by emphasizing the psychological and functional impact of facial deformities. Clarify why traditional methods such as autologous repair are often inadequate. Also, a context for osseointegrated implants should include a brief explanation of their origins and evolution from dental to craniofacial applications.
Better articulate the limitations of the current literature, such as small sample sizes and inconsistent findings regarding the effects of radiation. In this context, ensure that the study objective is clear and concise.
C. The Methods section should be reorganized into clear subsections (e.g., "Study Design," "Patient Selection," "Implant Procedure," "Complications," "Statistical Analysis") for readability and logical flow. Additional considerations - state how missing data (e.g., radiation dose information for six patients) were handled in the analysisÈ™ include the number of surgeons involved and their level of experience.
D. Results section: divide the section into subsections based on key aspects, such as "Patient Demographics," "Implant Outcomes," "Complications," and "Factors Influencing Success", etc.; use tables and figures to enhance clarity and avoid overly detailed descriptions in the text.
E. Divide the Discussion into clear subsections for ease of reading:
Summary of key findings points out the most important findings.
Literature comparison: relate the findings to existing research.
Clinical implications: discuss the implications for practice.
Limitations - there is already a paragraph on this issue
Future directions: suggest areas for further research
F. The Conclusions section should succinctly summarize the key findings of the study, highlight its clinical relevance, and end with a call to action or direction for future research. Currently, it effectively states the importance of osseointegrated implants but could benefit from more focus and impact.
G. The references are not written using MDPI style.
Author Response
Comment 1: Abstract - remove ”study design”; separate the purpose of the study from the methods. State the retrospective nature of the study early in the abstract. Include sample size and time frame concisely. Highlight key numerical results with concise language, such as implant success rates. Explicitly state the significance of results related to radiation and surgery.
Response: This has been edited and some of this was already present (sample size, time frame, implant success).
Comment 2: Introduction - reinforce this section by emphasizing the psychological and functional impact of facial deformities. Clarify why traditional methods such as autologous repair are often inadequate. Also, a context for osseointegrated implants should include a brief explanation of their origins and evolution from dental to craniofacial applications.
Better articulate the limitations of the current literature, such as small sample sizes and inconsistent findings regarding the effects of radiation. In this context, ensure that the study objective is clear and concise.
Response: The psychosocial impacts are imperative to touch on, and we do so heavily starting on line 233. Given the length already of the paper, we felt bringing up the idea in the intro was appropriate, and allowed for more in depth discussion in the discussion. Please see line 56,57 regarding prior research limitations.
Comment 3: The Methods section should be reorganized into clear subsections (e.g., "Study Design," "Patient Selection," "Implant Procedure," "Complications," "Statistical Analysis") for readability and logical flow. Additional considerations - state how missing data (e.g., radiation dose information for six patients) were handled in the analysisÈ™ include the number of surgeons involved and their level of experience.
Response: I am happy to edit the subsections of the paper, however this was the only review mentioning this and this was not in the author guide. Prior to editing, please confirm this is something that the journal would like and I will edit immediately. Also please see line 72 for number of surgeons.
Comment 4: Results section: divide the section into subsections based on key aspects, such as "Patient Demographics," "Implant Outcomes," "Complications," and "Factors Influencing Success", etc.; use tables and figures to enhance clarity and avoid overly detailed descriptions in the text.
Response: please see above comment about sub section reformatting in comment 3. Happy to change if this is ok by the journal.
Comment 5: Divide the Discussion into clear subsections for ease of reading:
Summary of key findings points out the most important findings.
Literature comparison: relate the findings to existing research.
Clinical implications: discuss the implications for practice.
Future directions: suggest areas for further research
Response: please see above comment about sub section reformatting in comment 3. Happy to change if this is ok by the journal. Added section about future directions and highlighted it in the latest draft.
Reviewer 4 Report
Comments and Suggestions for Authors
Defects in maxillo-facial area not only distort the functions of the biological systems, but also have devastating psychological effects and social impact. Autologous repairs and prostheses are the two possible options used to overcome the physiological, esthetic, and psychological complications from congenital malformations and defects from traumas and malignant diseases. As alloplasty may require multiple surgical procedures, facial prostheses appear to be an alternative in the repair of the defects in the facial area providing symmetrical, more esthetic and realistic appearance. Stability and retention of the restoration may be significantly improved if osseointegrated implants are used. Nevertheless, there are concerns about the implant placement and integration after possible bone alterations due to radiation therapy and surgical resection.
The study assessed the outcomes of auricular, nasal and orbital reconstruction performed by using osseointegrated implants for retention of prostheses.
The manuscript is well structured. The introduction provides detailed information about the possible treatments of maxillo-facial defects and the use of osseointegrated implants in the oral cavity and facial area. The materials and methods used, as well as the characteristics of the patients included in the study, are thoroughly described. The results concerning implant failure and inability to wear the final prosthesis are clearly presented. In the discussion section the results are analyzed and compared with the ones of other researchers. The limitations of the study are also discussed. A clear conclusion is stated. The references are relevant.
However, I have some recommendations:
- The abstract should begin with the background, not with the study design
- The quality of English must be improved – sentence starting on line 10
- A reference about the side effects of adhesives should be provided
- It would be interesting to provide information about the materials used for prostheses production
- Clearly state the level of significance – only p-values are provided in the result section
- The sentence on line 127: “There was a statistically insignificant difference in the number of patients who lost implants.” – please specify
- In table 4 it is not clear for which measurements the p-values are given
- It would be interesting to compare the failure rate of implants placed using one- and two-staged procedures
The manuscript would be of interest for the specialists working in the maxillo-facial area, and would contribute to the field of knowledge concerning the application of osseointegrated implants.
Author Response
Comment 1: the abstract should begin with the background, not with the study design
Response: This has been edited
Comment 2: The quality of English must be improved – sentence starting on line 10
Response: I was born speaking English and it is my only language. I'm not exactly sure which parts of the article the English should be improved as myself and the other authors (also native English speakers) have not identified specifics that need to be addressed.
Comment 3: A reference about the side effects of adhesives should be provided
Response: This has been added to the introduction.
Comment 4: It would be interesting to provide information about the materials used for prostheses production
Response: We can certainly discuss this further with our anaplastologist. I am only hesistent given the abundance of information already presented if this would deter from our outcomes objective. All the prosthesis were created by the same two people form the same company. I can add a statement regarding that or can get specifics if this is deemed to be vital to the paper.
Comment 5: Clearly state the level of significance – only p-values are provided in the result section
Response: This has been added and highlighted in the revised document.
Comment 6: The sentence on line 127: “There was a statistically insignificant difference in the number of patients who lost implants.” – please specify
Response: This line was redundant and confusing and therefore was removed. This is in addition to more explicit language regarding statistical significant in the same paragraph.
Comment 7: In table 4 it is not clear for which measurements the p-values are given
Response: This does look particularly abnormal when reformatted for the article journal. The p values are in general for the pair that is above them (ie for the first p value 0.188, this is referred to acute complications vs none in the radiated and non radiated patient). I did try to make the columns longer so they all fit on the line, but if this is accepted I can redo the table in a way that was found to be more pleasing.
Comment 8: It would be interesting to compare the failure rate of implants placed using one- and two-staged procedures
Response: We did consider this, however there were only 6 patients that had a two staged procedure. Given the clear recommendations on when to proceed with them, we decided to focus on other difference to compare. We could include another table comparing the difference if that would be fruitful. Table 3 includes the two patients who could not use their prosthesis, which does include information regarding one two-stager and one one-stager
Reviewer 5 Report
Comments and Suggestions for Authors
Dear Authors,
Thank You for a pleasure to read Your manuscript.
I have several comments and notes to improve Your article.
Title
Please, write here the type of study.
Abstract
Please, write separate accurate background and aim. It is not clear what radiation and why it affects results of reconstruction.
Key words
Please, add more, for example, radiation
Introduction
Please, add links for lines 38-43.
Please, add information for implants in reconstruction.
Please, re-praise the aim in the end of introduction with one point.
M&Ms
For ICD, please, write the number.
Lines with complications are more appropriate for results.
Statistics section is not enough, please, write it with details.
Procedure description is not clear especially You had patients with different localization for implants inserting. Please, write the main specific steps for all groups of patients.
Please, write in this section clear criteria for inclusion/ exclusion of patients.
Results
Table 1 describes characteristics of the patients it does not contains criteria for study inclusion.
Please, use statistics test for all counts where it is possible.
For table 4 You have no significant results, please, write this point in the text.
Also, You have not enough patients with secondary radiation for some conclusion. Also, for prior radiation You had no patients with orbit that means You could make conclusion only for nasal and ear implants.
Table 5 is more appropriate as text for comparison with Your results in discussion
section. Please, replace it and re-write.
Discussion
Figures are more appropriate for Materials or results, please, replace it.
Conclusion
Please, re-write this section according to Your study results.
Sincerely, Reviewer
Author Response
Comment 1: Title
Please, write here the type of study.
Response: Done
Comment 2: Abstract
Please, write separate accurate background and aim. It is not clear what radiation and why it affects results of reconstruction.
Response: Background subsection added. I do think the effects of radiation on healing are generally accepted given the poor wound healing and limited vascularity, I was worried that would be a bit much for the abstract, but happy to add if that is important.
Comment 3: Key words
Please, add more, for example, radiation
Response: done
Comment 4: Introduction
Please, add links for lines 38-43.
Please, add information for implants in reconstruction.
Please, re-praise the aim in the end of introduction with one point.
Response: Please see additions in the paper
Comment 5: M&Ms
For ICD, please, write the number.
Lines with complications are more appropriate for results.
Statistics section is not enough, please, write it with details.
Procedure description is not clear especially You had patients with different localization for implants inserting. Please, write the main specific steps for all groups of patients.
Please, write in this section clear criteria for inclusion/ exclusion of patients.
Response: Statistics have been further fledged out with statistical significance. The ICD 10 codes will add significant lines and this was the only reviewer requesting it. I am happy to add if felt this is necessary for the paper.
Comment 6: Results
Table 1 describes characteristics of the patients it does not contains criteria for study inclusion.
Please, use statistics test for all counts where it is possible.
For table 4 You have no significant results, please, write this point in the text.
Also, You have not enough patients with secondary radiation for some conclusion. Also, for prior radiation You had no patients with orbit that means You could make conclusion only for nasal and ear implants.
Table 5 is more appropriate as text for comparison with Your results in discussion
section. Please, replace it and re-write.
Response: Table 1 is just for patient demographics, it is not to discuss the inclusion criteria. Statistics in table 1 and 2 were not included as there were no significant differences, and it was difficult to compare even such limited samples for nose and namely orbit. In table 4, we do have significant results defined as p value < 0.05. I have edited the location of the discussion of table 5.
Comment 7:Discussion
Figures are more appropriate for Materials or results, please, replace it.
Response: thank you for this feedback, I did not format the paper that was by the journal.
Comment 8: Conclusion
Please, re-write this section according to Your study results.
Response: Is there something more specific about rewriting it? We used our results to guide the discussion and conclusion. Appreciate any further feedback you have
Round 2
Reviewer 3 Report
Comments and Suggestions for Authors
Dear authors,
Thank you for your resubmission and for addressing some of the previous comments. While I appreciate your efforts, several key recommendations have not been fully implemented and certain aspects still require revision to improve clarity, structure, and adherence to journal guidelines. I outline below the necessary improvements that should be addressed before final acceptance.
1. Abstract - the structure of the abstract still lacks a clear separation between the purpose of the study and the methods. The retrospective nature of the study should be explicitly stated at the beginning. Key numerical results, such as sample size and implant success rates, should be highlighted in a more concise manner.
2. Introduction -Although the psychosocial impact of facial deformities will be discussed later in the manuscript, a brief mention in the introduction is important to set the stage for the relevance of the study. The literature review should explicitly address sample size limitations and inconsistent findings regarding radiation effects.
Therefore the introduction should be slightly expanded to provide better context and to improve the transition to the rationale of the study.
3. Methods section - The section should be restructured into clear subsections to improve readability. Suggested subsections include Study design/ Patient selection/ Implant procedure/ Complications/ Statistical analysis
The handling of missing data should be explicitly stated (e.g. how the lack of radiation dose information for six patients was handled in the analysis).
The number of surgeons and their level of experience should be clearly stated.
Therefore, implement a structured format and provide details of how missing data were handled.
4. Results section - the text remains dense and would benefit from being broken down into structured subsections such as Patient demographics/ Implant results/ Complications/ Factors influencing success
The reliance on extensive textual descriptions should be minimized by using tables and figures.
5. Discussion section - The discussion lacks structured subsections, making it difficult to follow.
6. References and citations
The citation style and formatting of references do not conform to the rules of the journal.
References should follow the required style guide, including appropriate citation numbering and formatting.
Several comments were rejected based on the journal's formatting guidelines. However, restructuring sections for logical flow is beneficial regardless of journal-specific formatting preferences.
Significant improvements in structure, clarity and adherence to journal guidelines are still required. Please revise accordingly to improve the readability of the manuscript and its compliance with scientific publishing standards.
Author Response
Comment 1. Abstract - the structure of the abstract still lacks a clear separation between the purpose of the study and the methods. The retrospective nature of the study should be explicitly stated at the beginning. Key numerical results, such as sample size and implant success rates, should be highlighted in a more concise manner.
Response: the abstract has been edited with the study and methods with more clear separation. The retrospective nature is stated clearly. The key results have been highlighted.
Comment 2. Introduction -Although the psychosocial impact of facial deformities will be discussed later in the manuscript, a brief mention in the introduction is important to set the stage for the relevance of the study. The literature review should explicitly address sample size limitations and inconsistent findings regarding radiation effects.
Therefore the introduction should be slightly expanded to provide better context and to improve the transition to the rationale of the study.
Comment: Psychosocial impact was discussed briefly in 32, but now has been more extensively added. The limited sample size and inconsistent findings with radiation was already listed in lines 58-60, but I just made it more clear.
Comment 3. Methods section - The section should be restructured into clear subsections to improve readability. Suggested subsections include Study design/ Patient selection/ Implant procedure/ Complications/ Statistical analysis
The handling of missing data should be explicitly stated (e.g. how the lack of radiation dose information for six patients was handled in the analysis).
The number of surgeons and their level of experience should be clearly stated.
Therefore, implement a structured format and provide details of how missing data were handled.
Response: as stated in my last response, the number of surgeons is clearly stated. I added their experience. Subsections as requested were added.
Comment 4. Results section - the text remains dense and would benefit from being broken down into structured subsections such as Patient demographics/ Implant results/ Complications/ Factors influencing success
The reliance on extensive textual descriptions should be minimized by using tables and figures.
Response: subsections as requested were added. Textual descriptions have been thinned
Comment 5. Discussion section - The discussion lacks structured subsections, making it difficult to follow.
Response: subsections added as requested
Comment 6. References and citations
The citation style and formatting of references do not conform to the rules of the journal.
References should follow the required style guide, including appropriate citation numbering and formatting.
Several comments were rejected based on the journal's formatting guidelines. However, restructuring sections for logical flow is beneficial regardless of journal-specific formatting preferences.
Response: references have been formatted the first of the 3 MDPI reference list and citations guide.
Reviewer 5 Report
Comments and Suggestions for Authors
Dear Authors,
Thank You for Your corrections.
Sincerely, Reviewer
Comments on the Quality of English LanguagePlease, check English of article as it is not in my zone of expertise.
Sincerely, Reviewer
Author Response
Comment 1: Please, check English of article as it is not in my zone of expertise.
Response: English is my expertise along with my co-authors. We do not find any issues nor do the other reviewers comment on this. Thank you!